# Caregiving in Older Adults; Experiences and Attitudes toward Smart Technologies

**DOI:** 10.3390/jcm12051789

**Published:** 2023-02-23

**Authors:** Antoine Piau, Zara Steinmeyer, Nora Mattek, Allison Lindauer, Nicole Sharma, Nicole Bouranis, Katherine Wild, Jeffrey Kaye

**Affiliations:** 1Internal Medicine and Gerontology, University Hospital of Toulouse, Université Paul Sabatier, 31062 Toulouse, France; 2Oregon Center for Aging & Technology (ORCATECH), Oregon Health & Science University, Portland, OR 97239, USA

**Keywords:** assistive technology, in-home monitoring, online survey, social media, caregiving, older adults

## Abstract

(1) Background: The development of assistive technologies has become a key solution to reduce caregiver burden. The objective of this study was to survey caregivers on perceptions and beliefs about the future of modern technology in caregiving. (2) Methods: Demographics and clinical caregiver characteristics were collected via an online survey along with the perceptions and willingness to adopt technologies to support caregiving. Comparisons were made between those who considered themselves caregivers and those who never did. (3) Results: 398 responses (mean age 65) were analyzed. Health and caregiving status of the respondents (e.g., schedule of care) and of the care recipient were described. The perceptions and willingness to use technologies were generally positive without significant differences between those who ever considered themselves as caregivers and those who never did. The most valued features were the monitoring of falls (81%), medication use (78%), and changes in physical functioning (73%). For caregiving support, the greatest endorsements were reported for one-on-one options with similar scores for both online and in-person alternatives. Important concerns were expressed about privacy, obtrusiveness, and technological maturity. (4) Conclusions: Online surveys as a source of health information on caregiving may be an effective guide in developing care-assisting technologies receiving end users’ feedback. Caregiver experience, whether positive or negative, was correlated to health habits such as alcohol use or sleep. This study provides insight on caregivers’ needs and perceptions regarding caregiving according to their socio-demographic and health status.

## 1. Introduction

With the population ageing, the burden of chronic disease will increase [1] and so will the need for the caregiving support of family and friends. Up to 90% of in-home long-term care needed by adults is provided by unpaid caregivers [2,3] and usually involves assistance with basic (e.g., bathing, dressing) or instrumental activities of daily living (e.g., driving, medication management, helping with financial issues). While caregiving experience may have positive aspects [4], it may also compromise the caregiver’s physical and psychological health [1,5]. Zarit et al. proposed the following definition of caregiver burden as “the extent to which caregivers perceive that caregiving has had an adverse effect on their emotional, social, financial, physical, and spiritual functioning” [6]. As such, assistive technologies have been developed to support and help ease the burden on family caregivers. The solutions can range from educational websites to home assistance (fall detection device, connected health device) or robotic devices (robotic walking frame…) [7,8,9]. Their development is often not optimally pursued due to insufficient data on their efficacy and on the caregiver’s needs and perceptions of them. Moreover, most studies dealing with these considerations have not taken into account the caregiver’s social and health status as well as their experiences in providing such assistance, which could lead to the development of more personalized devices.

This study developed an online survey of the caregiver’s feedback on the use of information and communication technology supporting their roles. The primary objective was to obtain the caregiver’s perceptions and willingness to adopt assistive technologies. The second objective was to collect the socio-demographic and health status of caregivers.

## 2. Materials and Methods

### 2.1. Survey Development

Demographics and clinical characteristics of participants were collected via an online form. Consent to participate in the RITE (Research via Internet Technology and Experience) online cohort was obtained prior to participation [10]. A brief “Caregiving and Smart Home Technologies Survey” was developed based on a literature review [7,8,11,12,13,14,15,16,17,18]. Participants were asked about their caregiving history, their attitudes and perceptions concerning a selection of assistive technologies as well as their willingness to use them. The core of the survey was composed of questions with detailed sub-questions depending on answers (e.g., age of care recipient displayed only if the respondent had ever considered her/himself as an unpaid caregiver). Additional questions touched on hypothetical use cases to help the participants imagine themselves using the technology (Table 1). Participants also had the opportunity to provide free text comments.

### 2.2. Participants and Data Collection

The survey was developed using Qualtrics (Qualtrics^®^, Provo, UT, USA) survey software and administered as an online form through the RITE Program. The Oregon Center for Aging and Technology (ORCATECH) launched the RITE program in 2015. It was approved by the Oregon Health and Science University IRB (IRB00010237). RITE is designed to better understand people’s health needs, their perceptions of health-related technologies (and their willingness to use them), and how internet-based research may improve health care. RITE participants were recruited from the university’s health care system electronic medical record (EPIC EMR) and invited to participate as part of an online research cohort available over time to participate in surveys. Inclusion criteria were 18 years of age or older, owning and using a computer or smart device at least once per week, having a wired broad band Internet connection, and demonstrating English language. Only a sub sample from the RITE program responded to this survey.

In addition to episodic topical surveys (such as the survey reported here), all participants responded online every two weeks to a brief (<five minutes to complete) health and activity survey that provides information related to their current health and wellness status (e.g., mood, pain, illness, medication changes, visitors). Details of the RITE program are described at https://www.ohsu.edu/oregon-center-for-aging-and-technology/rite-study (accessed on 7 August 2022).

Consent to participate was obtained between August 2015 and November 2016 (IRB #10237). Participants received an email with a link to the present survey and were asked to complete it. Between 16 January 2017 and 23 February 2017, 411 volunteers took this one-time survey.

### 2.3. Survey Analysis

Demographics, clinical characteristics, and survey responses were summarized for the overall cohort. Comparisons were made between those who ever considered themselves caregivers and those who never did. Differences between the two groups were assessed using t tests or the Wilcoxon rank sum test for continuous variables or Pearson’s chi-square test of Fisher’s exact test for categorical variables as appropriate. All analyses were performed using SAS 9.4 software (SAS Institute, Inc., Cary, NC, USA).

## 3. Results

### 3.1. Demographic and Health Characteristics of Respondents

Of 411 volunteers that responded to the survey (from 773 who were sent the survey), 398 answered our primary question “Have you ever considered yourself an unpaid caregiver for an adult over the age of 30?” The mean age was 65 years (range 28–95) and 54% were female. Survey respondents’ characteristics are presented in Table 2. Half of the respondents (49%) had caregiving experience. Those who have ever considered themselves an unpaid caregiver were more likely to be women (68% vs. 41%, *p* < 0.0001) and to live alone (26% vs. 15%, *p* < 0.01). Those who had been a caregiver also had more health conditions (5.6 vs. 4.7, *p* < 0.001). Among all respondents, 34% self-reported depression in the last two years, 30% reported using medication or alcohol to sleep, 13% of respondents reported having three or more alcoholic drinks every day, and 17% reported fair or very poor sleep quality. The mean number of current close friends was 6.0 (SD = 5.3, range 0–40). Among those who have been a caregiver, they responded on average 2.6 on a scale from 0 to 5 (SD = 1.2) to the question “How difficult was it for you to help with those duties?” Participants also reported how much strain they felt as a caregiver. Mean physical strain was rated as 1.9 (SD = 1.3), emotional stress as 3.1 (SD = 1.4), and financial strain as 1.8 (SD = 1.5) out of 5. More than half of the caregivers (56%) felt they had no choice in taking on the caregiving responsibility. Almost all respondents (99%) reported using a computer for 5 years or more and 79% rated their confidence level using computers as high (4 or 5 out of rating from 1 = “total lack of confidence” to 5 = “extremely confident”). Most respondents (85%) owned cell phones; among them, 75% had smartphones.

### 3.2. Caregiver’s Experience

Of the respondents, 49% had been an unpaid caregiver for an adult over the age of 30, and a majority for several years (61%; 16% for less than 6 months). The age of the care recipient ranged from 32 to 102 years. Care recipients were 57% female, and most of them were a mother or mother-in-law (66.5%) or a wife/partner (17%). When the caregiver was a male, the relationship was primarily a father or father-in-law (38%) or a husband or a partner (35%). The care recipients who did not live in their caregiver’s home lived alone for 51%, of whom 69% resided in their own home. Care scenarios are summarized in the Appendix A Table A1. The top underlying etiologies for providing care were Alzheimer’s disease and other dementia (21%) and cancer (18%). Sensory disorders (2.5%) or other disease entities (e.g., Parkinson’s disease, 2%; stroke, 3%) were not frequently reported as reasons for care provision.

### 3.3. Attitudes about In-Home ‘Smart’ Technologies

Participant attitudes and perceptions about assistive technologies are summarized in Table 3 There were no differences between the groups. The use of a voice-activated device providing assistance was the technology most agreed upon to consider for use. Less than a third were concerned about privacy breaches when using these technologies. When asked who they thought should have access to the care recipient’s data, 84% answered “their caregivers”, 77% “the person him/herself”, 76% “their doctor”, 59% “their family members”. If the assistive technology was used for themselves, respondents said they would allow the following to access their data: “their doctor” (66%), “their spouse” (69%), and “their family members” (58%).

Potential positive aspects of smart home technologies were highlighted in the free text comments. Among them, remote monitoring to “check in on someone” was highlighted (“you could monitor any problems that may arise while you were away”) as well as the limits of current tools (e.g., falls, “she will not wear the alarm medallion”). Assistance in mobility (“a device that could help a patient from sitting position to rising to his feet”) was also pointed out as an important issue. Other caregiver comments not covered in the survey were the importance of addressing loneliness, exercise motivation, and food management with health devices. The need to educate people in using the already available technologies was another notable comment.

Several concerns were also expressed in the free comments about safety (“threat of being hacked”, “security is lacking on most of these Internet-of-Things devices”), privacy (“doubt mother would accept the intrusion”, “I consider this an invasion of privacy”), false alarms (“would it give accurate information or send distress signals when not needed”), maturity (“technologies would have to be much more reliable than they are presently”), usability (“my parents have a hard time just using their smartphones”), additional costs (“waste of money when the caregiver can easily know all these issues”), and the threat of replacing direct human contact (“no computer or system replicate human interaction”).

### 3.4. Use Cases

Answers to questions related to hypothetical use cases are detailed in Table 4. For use case 1, respondents consistently thought that the following events or activities would be the most helpful to check in on a person they are caring for: falls or risk of falling (81%), medication use (78%), changes in physical functioning (73%), overall activity level (61%), and change in memory (57%). For use case 2, concerning caregiving support, the greatest interest was found in the one-on-one options with very similar scores for both online and in-person alternatives. Respondents declared that they were familiar with this kind of technology (82%) and if they had to choose one of the online options, their first choice would be the online, one-on-one arrangement. For use case 3, the concerns about participating in an online peer-to-peer support organization were mainly privacy (70%), security (65%), cost (62%), and obtrusiveness (55%). Some caregivers freely expressed concerns on the possibility of misuse (“access by health insurer to raise my costs”), reliability and false alarms (“should be well integrated and validated before deployment”), and the willingness to stay “in control” (“I would not be seen as the coordinator of my care”).

## 4. Discussion

### 4.1. Main Results

The RITE survey demonstrated the feasibility and potential utility of online cohorts to collect information on technology perceptions, as well as detailed health and social profiles of large samples of participants. Beyond basic health data, highly relevant information was collected regarding caregiving issues such as depression, sleep issues, medication and alcohol habits, social interactions, and perceived strain [1]. Moreover, detailed data about the caregiving status of respondents (e.g., care schedule,) and of care recipients (e.g., living arrangement) were also reported.

The overall responses were very positive concerning the willingness to use health technologies = (e.g., smart-speaker device). Surprisingly, the robotic pet was among the least desired technology (39% of positive responses), despite the large research popularity of this kind of solution [19,20,21,22]. Among the potential positive aspects of smart home technologies, remote monitoring was consistently brought to the forefront both in the questions–answers and the free comments, especially for falls, medications use, and change in functioning. Assistance in mobility was also highlighted.

Interestingly, most of the respondents (71%) thought smart technologies would be helpful in the home of someone they are caring for, but only 53% would consider using a smart system for themselves in their own home. We also found a large adherence for the remote monitoring of others but only 40% of the respondents would accept it for themselves. This could be explained by the serious concerns expressed about safety, privacy, obtrusiveness, and technological maturity.

The possibility of the replacement of human interaction by technology was also expressed. However, concerning caregiving support, the scores were not driven by the use of technology or not, but rather by the preference of one-on-one interactions over group alternatives. Moreover, an interesting comment pointed out the possibility of using health technologies to facilitate human interactions, rather than replace them. Such duality in perception has already been described [11].

### 4.2. Findings in the Context of Previous Research

The profile of our caregivers is concordant with the literature. The majority of caregivers are women who take care of a relative and spend several hours a day providing care, with 20% spending more than 40 h per week [1,2,3]. We identified several studies dealing with perceptions of technologies to support caregiving. However, they included small samples of multiple stakeholders (patients, health professionals, and caregivers), focused on a specific disease or technology [12,13,14,15,16], and did not address the perception of technology use for the care recipient and for the caregiver. New technologies were generally perceived positively [12,13,14,15,16] although there was some concern about security issues [12,13,16]. Darragh et al.’s study focused on the caregivers of patients with mild cognitive impairment and reported that monitoring health outcomes such as falls, abnormal inactivity, change in cognition, behavior, and mood was an important issue [23].

Frequently cited resources when it comes to technology use and perception are prior surveys on topics such as “Older adults and technology use” [17] or “Americans attitudes toward robots as caregivers” [18]. These surveys provide results without much description of the participant sample (e.g., health status, experience as a patient or caregiver), making it difficult to cross-reference the results with this study. The American Association of Retired Persons (AARP) [2] regularly provides detailed data on the demographic characteristics of a large panel of caregivers and care recipients in the United States (*n* = 1248), the caregiver’s situation (e.g., intensity of care), and how caregiving affects caregivers’ strain. The 2015 AARP study is the first to use online data collection (interviews) instead of the traditional landline-only telephone study. However, they did not address the technological perception issue.

### 4.3. Limitations

As with all online surveys, there are limitations with regard to the sample. By definition, all respondents were online; the responses did not cover those who are not online or are not willing to answer surveys online. Therefore, this may reduce generalizability to participants who are less tech savvy. In addition, the responses were associated with individuals who had received care at Oregon Health and Science University and had a record in the OHSU EMR system. They were, thus, largely Oregonians with the majority residing in the Portland, Oregon metropolitan area. Respondents were mostly white, with a high education level and good computer literacy (87% had been using a computer for at least 5 years and most reported being confident with its use). One of the advantages of communication technologies is the possibility to overcome geographical and social barriers and, thus, future studies may address more heterogeneous populations. Moreover, data collection was obtained in 2017, so experiences and attitudes toward smart technologies may differ from today. Finally, the study population was relatively young (mean age, 65). Older individuals may encounter unique challenges in the use of everyday technologies (e.g., related to sensory and cognitive decrements) which may change their perception and willingness to use these technologies [24].

## 5. Conclusions

The aim of this study was to survey caregivers on their perceptions and willingness to adopt assistive technologies, and to collect data on their socio-demographic and health status. The use of an online survey allowed us to reach a large panel with almost 400 respondents. Perceptions and willingness to use technology were generally positive, with no significant differences between those who had previously considered themselves caregivers and those who never have. Interestingly, in terms of support for caregivers, individual options received the most approval, with similar scores for online and face-to-face options. There were significant concerns about privacy, intrusiveness, and technological maturity.

## Figures and Tables

**Table 1 jcm-12-01789-t001:** Questions related to hypothetical use cases.

Use case 1, acting as care provider: “Imagine a set of ‘smart technologies’ that can be placed in a home to monitor certain activities or behaviors, can you rate the importance of different monitoring targets?”
Use case 2, acting both as care provider and/or care recipient: “Imagine there is a neighborhood organization that provides its members with peer-to-peer support services (e.g., transportation to the doctor). You would use a web-portal or a smart phone app to know when a user might need help or to request help for yourself.”
Use case 3, acting as care provider: “Imagine you are given an option for support in your caregiving role. How do you feel about these different options? Online one-on-one support: live videoconferencing with a clinician; online support group: live videoconferencing with a group of 6 peers; in person one-on-one support: meet at the clinician’s office; in-person support group: meet at an office setting near you.”

**Table 2 jcm-12-01789-t002:** Characteristics among those who have and who have not ever considered themselves an unpaid caregiver.

	Total (*n* = 398)	Ever Caregiver (*n* = 196)	Never Caregiver (*n* = 202)	*p*
Age, years (SD)	65.0 (11.2)	65.2 (10.1)	64.8 (12.2)	0.76
Gender (% female)	54	68	41 **	<0.00
Education (% college graduate)	81	81	81	0.85
Race (% white)	93	92	94	0.37
Living alone (%)	20	26	15 *	<0.01
Number of current health conditions (SD)	5.1 (2.5)	5.6 (2.6)	4.7 (2.4) ***	<0.00
Number of current medications (last two weeks) (SD)	4.1 (3.4)	4.1 (3.4)	4.1 (3.5)	0.90
Subjective Memory loss (%)	25	23	27	0.41

* *p* < 0.01; ** *p* < 0.0001; ****p* < 0.001.

**Table 3 jcm-12-01789-t003:** Main attitudes about in-home smart technologies among those who have and have not ever considered themselves as an un-paid caregiver.

	Total (*n* = 398)	Caregiver (*n* = 196)	Never Caregiver (*n* = 202)
Would technology be helpful in the home of someone you are caring for? (Yes, %)	71	67	75
Would you consider using a smart system for yourself in your own home? (Yes, %)	53	49	56
Would you consider using a remotely driven assistive smart-speaker device for yourself or a loved one? (Yes, %)	79	77	80
Would you consider using a remotely controlled assistant robot—“video chat or check-in on wheels”—for yourself or a loved one? (Yes, %)	69	65	73
Would you consider using a robotic device to help move the care recipient from the bed to another location ? (Yes, %)	69	65	74
Would you consider the idea to use a robotic pet for yourself or a loved one? (Yes, %)	39	40	38
Are you concerned about privacy in relation to having these technologies in your home? (Yes, %)	29	27	32
Are you concerned about personal data getting to people or organizations that do not have a right to it? (Yes, %)	46	47	45

Note. There were no statistical differences between groups.

**Table 4 jcm-12-01789-t004:** Use cases related to technology use to support caregiving.

**Use case 1, acting as care provider:** “Imagine a set of ‘smart’ home monitoring technologies. How important these targets would be?”	**Very Important (%)**	**Somewhat Important (%)**	**Not Very Important (%)**	**Not Important at All (%)**
Falls or the risk of falling	73	21	4	3
Medication use	73	19	4	4
Changes in physical functioning	63	28	6	3
Changes in memory	49	37	9	5
Blood pressure or heart rate	45	45	5	5
Mobility such as movements around the house	42	42	10	5
Overall level of activity	37	46	12	5
Cooking	30	39	21	10
Nighttime activity	30	42	21	7
Time out of home	26	39	25	10
Visitors	25	38	28	9
Bathroom activity	23	51	19	7
Change in frequency/difficulty in using a computer	22	39	27	11
**Use case 2, acting as care provider and/or care recipient:** “Imagine an online web-portal/smart phone app organization to propose your help or to request help for yourself. Would you be willing to…?”		**Yes (%)**	**Maybe (%)**	**No (%)**
Join such a peer-to-peer organization?		55	35	10
Give care to fellow members?		44	50	6
Receive care from fellow members?		38	58	4
Learn how to use a smart home system to use the system?		62	30	8
Have a smart home system to check in on and help you?		40	46	14
**Use case 3, acting as care provider:** “How do you feel about these different support options in your caregiving role?”	**Very interested (%)**	**Interested (%)**	**Neutral (%)**	**Not interested (%)**
One-on-one live videoconferencing with a clinician	16	33	25	26
Group online live videoconferencing	5	25	29	41
In person one-on-one at the clinician’s office	15	36	28	21
In person group at an office setting near you	9	21	31	38

## Data Availability

The authors confirm that the data supporting the findings of this study are available upon reasonable request.

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
