# Peer review of "Caregiving in Older Adults; Experiences and Attitudes toward Smart Technologies"

_jcm, 2023, doi:10.3390/jcm12051789_

Round 1

Reviewer 1 Report

The topic of this study is interesting and relevant, but I have some concerns about the study goal and the study method.

-      -          My main concern is that the data are collected almost six years ago, in the first months of 2017. Especially in the area of smart technologies, technological advances are rapid. So, it may be that answers to the questions about the attitudes towards in-home smart technologies would be different now.

-          The goal of the study is not entirely clear to me. The primary goal seems to be older adults’ experience and attitude towards smart technologies. But it was also studied whether caregiver experience is related to health habits as alcohol use and. Although I do understand that these two topics may be related - smart technologies may reduce the burden on caregivers, and this burden may be reflected by their health habits -, this relation is not clearly described or explained. And a third study goal seems to be to investigate to what extent online surveys are an effective tool to collect health information on caregiving. While reading the article, this seemed to be a less important goal. However, the information in section 5, the conclusions, is almost exclusively related to this goal. I would advise the authors to clarify what the main and secondary study goals were.

-          Related to my previous remark; I am not sure why the long Table 3 is included. Although it may be interesting to learn more about the specific caregiving experiences of the respondents, this information does not seem to be directly related to the study goals. The information is also not related to the other results, although I can imagine it may be informative to relate results about, for example, the main reason for care assistance to the acceptance of certain smart technologies.

-      It is unclear to me why use case 2 about providing support services (this use case is later presented as use case 3 in Table 5) focuses on the respondents as care providers and/or care recipients. Why have the authors also focused on receiving care, and why only in this use case and not in use case 1?

-          As the authors explain, it is a serious limitation that all respondents were online. So, we do not know how elderly people who are less tech savvy, and who are not able (or willing) to answer online surveys think about smart technologies. It may be expected that their opinion differs from this group of elderly people with good computer literacy.

Author Response

Reviewer 1 :

  • As underlined by reviewer 1, this data was collected in 2015. However, these results remain original and are useful for further research on this matter and may be of greatest interest to the readers of JCM.

  • As advised by Reviewer 1 we have clarified the study goals in the manuscript (Line 52-54) "

    The primary objective was to  obtain caregiver’s perceptions and willingness to adopt assistive technologies. The second objective was to collect  socio-demographic and health status of caregivers."

  • Indeed, we considered the table 3 interesting as it is related to caregiving experience of the respondents. However, if the reviewer 1 wishes that this table is listed in the appendix we will change its position.

  • As pointed out by Reviewer 1, we have corrected the error in the use case numbers.

Moreover, we have chose Use case 2 as care provider and/or care recepient as it may help understand the perspective of both point of views and adapt  the demands when developing communication technologies. It seems difficult  to not integrate both point of views in this type of research.The study is mainly about caregivers opinions and perceptions on smart technologies, as such the majority of use cases are on the care providers perspective, we only added one use case containing both.

  • As pointed out by Reviewer 1, indeed one of the limitations of this article is that all respondents are online. We will add this limitation in the study limitations section (line 236). However, more and more older adults are tech users and this trend will continue to grow in the next years. So we don’t perceive this as a serious limitation.

Moreover, online surveys should be perceived as complementary to traditionnal surveys and not a substitute.

Reviewer 2 Report

First of all, I would like to thank you for the opportunity to review the paper entitled:Caregiving in older adults; experience and attitude towards smart technologies. In this research, authors evaluate caregivers needs and perceptions regarding caregiving according to their socio-demographic and health status. In general, the paper has an interesting and relevant topic for researchers and caregivers, and the written level of English is very good and easy to read.

However, the methodology and main results need to be rewritten. With your permission I recommend the following:

Line 58: This RITE platform (Research via Internet Technology and Experience) was developed for the caregivers? This information is not clear.

Line 68: this table was a good idea for the manuscript. The questions are all relevant for the study.

Line 69: Do you consider any inclusion - exclusion criteria for the participants?

Line 78: This two weeks evaluation was also insert in the study?

Line 85: Did you made any a priori analysis? Did you consider the type 2 error? There is any information about the study design? The results are significant, but about their effect? For a better understanding of the results, you should insert the descriptive statistics for all the variables from the study, like this will be easily for other researchers to replicate your study and the most import to involve other groups from other cultures.

Line 88: Please insert the report for t tests; Wilcoxon rank sum test  and chi-square test of Fisher’s and also effect size for each analysis.

Line 96: How about the range of the age? There is any differences of the perception according age?

Line 112: Most of 75% participants have two phones?

This study is very interesting from the perspective of caregivers' opinions about their perceptions. Is there a questionnaire on caregivers' perceptions? The questions bring to the researchers' knowledge another "vision" from which the caregivers can be observed. I think that by reporting all the results, highlighting the limits regarding the target group and promoting clear directions for those who want to use this questionnaire, they will improve your extraordinary work.

Author Response

Dear Reviewer 2, Thankyou for your thoughtful and interesting remarks.

Line 58 : The RITE program was not developed for caregivers, it is a web based survey collecting data from participants about their opinions and experiences with the intersection between technology and healthcare (tinyurl.com/38p4ez56).  This is a link to the RITE study website for further information: https://www.ohsu.edu/oregon-center-for-aging-and-technology/rite-study.

The RITE participants receive baseline surveys collecting self-reported medical histories and demographic metrics, annual surveys to update this information, bi-weekly surveys to assess health and important events in their life, as well as quarterly surveys (cross-sectional in nature) to assess specific research topics of interest. Like the survey reported here. We have detailed this in the methods section for more clarity.

  • Line 69 : As suggested b y Reviewer 2 we have added inclusion and exclusion criteria for the RITE program. We have added these criteria in the methods section.

  • Line 78 :The two weeks evaluation was not included in this study.

  • Line 85: This is a topical survey of a convenience sample of volunteers interested in providing their feelings and experience on a range of healthcare and technology topics. Therefore no a priori hypotheses, we did not consider type 2 error as it is a self-report study.

This ishe link to the RITE study website: https://www.ohsu.edu/oregon-center-for-aging-and-technology/rite-study. for further information.

This article resumes the RITE study : The survey for memory, attention, and reaction time (SMART): Preliminary normative online panel data and user attitudes for a brief web-based cognitive performance measure - PubMed (nih.gov)

We chose to not include all the other variables because there would be too much data not necessarily relevant for this study.

RITE is unique because it uses online surveys to collect data from thousands of participants of various ages and backgrounds. Participants answer questions about healthcare, technology and what role technology plays in their health and well-being. Instead of traveling to a clinic, participants can contribute to research from the comfort of their home, on a more regular and real-time basis.

  • Line 88 : We have added as suggested by Reveiwer 2 , the results of the bivariate analysis in Table 1.

  • Line 96 : Mean age and range age is noted in the results (line 101)

Perceptions according to age were not studied in this manuscript.

  • Line 112 : As suggested by Reviewer 2, we have modified the phrase for more clarity : “ Most respondents (85%) owned cell phones among them 75% had smartphones.

Round 2

Reviewer 1 Report

I’d like to thank the authors for their response to my comments. I think the article has improved by replacing table 3 and by correcting the error in the use case numbers.

With regard to my other comments, I still have some questions and doubts:

-          The data were collected in 2017 (according to the article, section 2.2) or in 2015 (according to the authors in their response to my comment). Although I do agree with the authors that the data remain original and although I do understand that the data may be useful for further research, I think the authors should reflect on the possible consequences of publishing research on data that were collected 6 or 8 years ago. I think that it may very well be that answers to the questions about the attitudes towards in-home smart technologies would be different now and I hope the authors can reflect on this.

-          The two goals of the study are clear to me now, but it is still a bit unclear why the conclusions section seems to be about a third – not explicitly mentioned – goal: investigate to what extent online surveys are an effective tool to collect health information on caregiving. I would have expected a conclusion about the two goals that were presented in the introduction.

Author Response

Dear Reviewer

Thankyou for your time and constructive comments.

  • Comment 1: Indeed, the data for this study was collected in 2017 as noted section 2.2. Consent to participate in the RITE study was obtained in 2015. I apologise for not being clear enough in my first response.
    We have added this phrase: in the limits of this manuscript, as we acknowledge that the same study today may present different responses. line 246-247
  • Comment 2: We have modified the conclusion so the two goals are clearly represented. line 253-261

Thankyou for your review